# Projection Domain Metal Artifact Reduction in Computed Tomography using Conditional Generative Adversarial Networks

**Nele Blum**[1]                                                      BLUM@IMT.UNI-LUEBECK.DE
[1] *Universität zu Lübeck, Ratzeburger Allee 160, 23562 Lübeck*

**Thorsten M. Buzug**[1,2]                          THORSTEN.BUZUG@IMTE.FRAUNHOFER.DE
**Maik Stille**[1,2]                                      MAIK.STILLE@IMTE.FRAUNHOFER.DE
[2] *Fraunhofer IMTE, Mönkhofer Weg 239 a 23562 Luebeck*

**Editors:** Under Review for MIDL 2021

## Abstract

High-density objects in the field of view, still remain one of the major challenges in CT image reconstruction. They cause artifacts in the image, which degrade the quality and the diagnostic value of the image. Standard approaches for metal artifact reduction are often unable to correct these artifacts sufficiently or introduce new artifacts. In this work, a new deep learning approach for the reduction of metal artifacts in CT images is proposed using a Generative Adversarial Network. A generator network is applied directly to the projection data corrupted by the metal objects to learn the corrected data. In addition, a second network, the discriminator, is used to evaluate the quality of the learned data. The results of the trained generator network show that most of the data could be reasonably replaced by the network, reducing the artifacts in the reconstructed image.

**Keywords:** CT, MAR, deep learning, cGAN

## 1. Introduction

In computed tomography (CT), artifacts are particularly apparent when metal objects are present in the field of view. They manifest as dark and bright streaks in the image and degrade the diagnostic value of the image. Since the first publications on metal artifact reduction (MAR) (Glover and Pelc, 1981), various MAR methods have been introduced. Sinogram completion methods treat the projection data affected by metal objects as missing data, which are often replaced by an interpolation (Veldkamp et al., 2010). Still, a major drawback is the loss of information in the metal trace, especially in inhomogeneous image regions. Recently, neuronal networks are used to fill in the missing data (Ghani and Karl, 2019; Peng et al., 2020). An essential part of successful training is the used loss function. Many loss functions are designed to optimize certain quantifiable image parameters. However, selecting image properties to define a good image quality is often non-trival. One possible strategy is to create another network that evaluates the image quality. In addition to the so-called generator network, which generates the improved projection data, a second network is constructed, the discriminator, which is trained to distinguish real sinogram from artificial data.

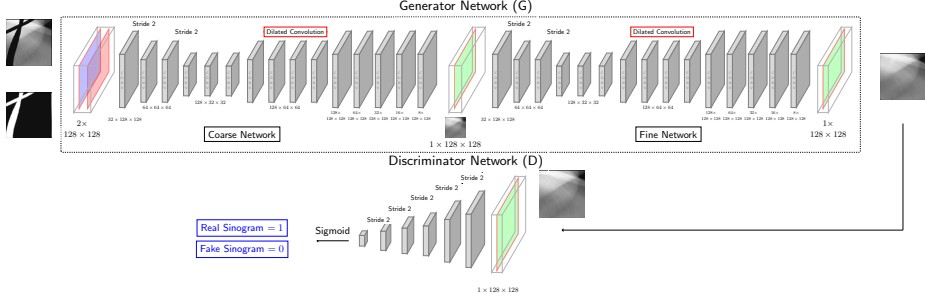

Figure 1: Schematic representation of the used network architectures.

## 2. Methods

To train the networks, an adversarial loss function is used in which the generator and discriminator compete against each other. For training, the loss function is minimized by the generator $G$ and maximized by the discriminator $D$. The objective function

$$G^* = \arg \min_G \max_D L_{\text{GAN}}(G, D) + \lambda \mathbb{E}_{x,x_{\text{ref}}}[\|(x_{\text{ref}} - G(x))_{\text{local}}\|_1 + \|(x_{\text{ref}} - G(x))_{\text{global}}\|_1] \quad (1)$$

with

$$L_{\text{GAN}}(G, D) = \mathbb{E}_{x,x_{\text{ref}}}[\log(D(x, x_{\text{ref}}))] + \mathbb{E}_x[1 - D(x_{\text{ref}}, G(x))] \quad (2)$$

is based on a Min-Max game between both networks, over the expectation values $\mathbb{E}$ of the training pairs $x$ and $x_{\text{ref}}$. To stabilize the training process, the loss is extended by a L1-loss in the local mask region and the global image patch weighted by the parameter $\lambda$. The generator receives the incomplete sinogram data and the corresponding metal trace as an input. Using a two-stage architecture consisting of a coarse and a fine network (Yu et al., 2018), the generator generates an output image that is used as an input to the discriminator. The discriminator, a binary classification network, assigns a probability between $[0, 1]$, indicating whether the data are real or artificially generated (Figure 1).

## 3. Results and Experiments

Simulated data from a software phantom (Segars et al., 2008) with 120 varying parameter settings were used to generate the data, which were divided into training (90), validation (15), and test (15) data. Objects were inserted at random positions in the phantom and metal affected projection data were generated using simulations for 3D cone-beam CT. In the image domain, the metal objects were segmented and projected forward to obtain the metal trace that is used to remove the metal corrupted data. To generate data with a higher variation, image sections of $128 \times 128$ were used instead of full sinograms. Training was performed with 183 000 training pairs. For validation, the Mean Squared Error (MSE) between the reconstructed images and the ground truth was calculated on the test data set. The network was able to learn most of the missing structures, but some streak artifacts remain visible in the reconstructed image (Figure 2). Compared to linear interpolation, the average MSE value overall test images is lower with $5.5 * 10^{-8}$ versus $6.2 * 10^{-8}$ .

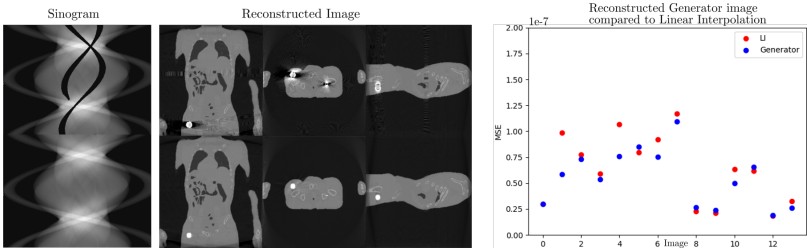

Figure 2: Data before (top) and after application of the network (below).

## 4. Conclusion

The results demonstrate that the generator is able to reduce a vast number of artifacts in the sinogram. For the following steps, the networks should be trained on real clinical data. Instead of using the data with the missing metal trace, the corrupted projection data should be used as the input of the generator network. The goal is for the network to be able to use the information from the metal trace to avoid introducing false image structures.

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
