# OpenReview forum: "Projection Domain Metal Artifact Reduction in Computed Tomography using Conditional Generative Adversarial Networks"
_MIDL.io/2021/Conference/Short — MIDL 2021 Poster_

### Official Review · Reviewer_6VjA · 2021-04-29

**Confidence:** 4
**Final Rating:** 3

**Summary:**

This work applied a conditional GAN with a two-stage generator to reduce the metal artifacts in CT images. The method was validated on a synthetic dataset in which the artifacts were also simulated. The experimental results showed the method was able to reduce a vast number of artifacts in the sinogram and achieved lower MSE than the linear interpolation.

**Strengths:**

- Well written and easy to follow.
- The idea to apply related image inpainting techniques for metal artifact reduction is interesting, given the two problems are similar.
- The experiment design is reasonable. I particularly appreciate that the authors split the dataset into training, validation, and test set based on the 120 source synthetic images, by which to avoid information leakage.


**Weaknesses:**

- The proposed framework is mainly based on the one from Yu et al, which deals with image inpainting, therefore the novelty of this work is discounted.
- Despite being a short paper, the validation was still not sufficient.


**Deanonymize Review:**

no

**Detailed Comments:**

- The proposed method compared only with the linear interpolation (LI), which was not a strong baseline. It is not very surprising that the proposed method, as a learning-based algorithm, can outperform the LI.
- The images with artifact for training and testing are synthetic. I am not sure whether the distributions of the generated artifacts in feature space can highly resemble those of realistic cases. The validation on a real dataset may be necessary.
- In equation (1), if L1-loss is used pixel-wisely, then the local reconstruction loss has already been included in the global reconstruction loss. The local reconstruction loss is therefore redundant, unless the authors see such a design as a way to give the local reconstruction loss a large weight.

**Justification Of The Rating:**

Given the problem context in this paper is similar to that of image inpainting, applying related techniques from image inpainting to reduce metal artifacts in CT images is sensible. While the work is weakened by the insufficient validation, it is a good attempt to introduce the conditional GAN for this problem.

**Paper Type:**

both

**Special Issue:**

no

---

### Official Review · Reviewer_RcRB · 2021-04-30

**Confidence:** 4
**Final Rating:** 3

**Summary:**

The authors proposed GANs for reducing the distortion caused by high density metal artifacts while reconstructing CT images. The standard GAN loss is extended by a L1 norm to ensure a stable training. The proposed method was evaluated on synthetic images with promising early results in terms of reduced artefacts.

**Strengths:**

1.	The Clinical motivation of the paper is clear.
2.	The paper is well-written and easy to follow.
3.	The illustrations are adequate to appreciate the problem as well as the method designed to solve it.


**Weaknesses:**

1.	Albeit a short paper, I expected to see some qualitative results on artefact removal from real images. Since that is not there, I am a bit uncertain about believing the authors’ claims. There are many articles that describe the simulation to real generalization problem in medical imaging. I expect a GAN-based approach would face similar problems.
2.	How to measure the usefulness of GAN based reconstruction. Visual inspection or L1-norm comparison between interpolated images is a rather simplistic approach. I want to see some thoughtful reflection here.
3.	Finally, the last part of the first paragraph of introduction contain information that is rather verbose. For the sake of brevity, please shorten significantly and just cite the most popular GAN reviews (https://www.sciencedirect.com/science/article/abs/pii/S1361841518308430, https://www.sciencedirect.com/science/article/pii/S0933365719311510).


**Deanonymize Review:**

no

**Justification Of The Rating:**

While the clinical motivation is strong, the authors need to address my main concerns before I can recommend for publication.

Changes in Final version
1.	Preferably some qualitative results on the performance on real images. If that is not possible, a few lines describing the potential shortcoming of the validation on synthetic images.
2.	A few lines on possible ways to measure the usefulness of GAN based reconstruction.
3.	You can remove details of GANs from introduction, by citing the two review articles of GAN that are widely cited. Use that space to address the previous points.


**Paper Type:**

validation/application paper

**Special Issue:**

no

---

### Meta-Review · Program_Chairs · 2021-05-06

**Recommendation:** Accept (Poster)
**Confidence:** 5

**Metareview:**

Both reviewers support acceptance. Authors are suggested to address reviewer comments in final version.

---

### Decision · Program_Chairs · 2021-05-11

Accept (Poster)